# Investigation of the impacts of climate change and rising temperature on food poisoning cases in Malaysia

Noor Artika Hassan[1,2,3,4]*, Jamal Hisham Hashim[3,4,5], Sharifa Ezat Wan Puteh[3], Wan Rozita Wan Mahiyuddin[6], Mohd Syazwan Faisal Mohd[7], Shazlyn Milleana Shaharudin[8], Edre Mohammad Aidid[1], Isnizam Sapuan[1]

1 Department of Community Medicine, Kulliyyah of Medicine, International Islamic University Malaysia, Kuantan, Pahang, Malaysia, 2 IIUM Health, Safety, Environment Unit, Kulliyyah of Medicine, International Islamic University Malaysia, Kuantan, Pahang, Malaysia, 3 Department of Community Health, Faculty of Medicine, UKM Medical Centre, Kuala Lumpur, Malaysia, 4 United Nations University-International Institute for Global Health, UKM Medical Centre, Bandar Tun Razak, Kuala Lumpur, Malaysia, 5 Department of Environmental Health and Occupational Safety, Faculty of Health Sciences, Universiti Selangor, Shah Alam, Selangor, Malaysia, 6 Institute for Medical Research, Ministry of Health Malaysia, Selangor, Malaysia, 7 National Hydraulic Research Institute of Malaysia, Ministry of Environment, Seri Kembangan, Selangor, Malaysia, 8 Department of Mathematics, Faculty of Science and Mathematics, Universiti Pendidikan Sultan Idris, Tanjong Malim, Malaysia

* noor_artika@iium.edu.my

**Data Availability Statement:** The confidential raw data is only for the Malaysian Ministry of Health Data for Food Poisoning. In Malaysia, all the diseases data are not accessible to the public. The

## Abstract

This study is an attempt to investigate climate-induced increases in morbidity rates of food poisoning cases. Monthly food poisoning cases, average monthly meteorological data, and population data from 2004 to 2014 were obtained from the Malaysian Ministry of Health, Malaysian Meteorological Department, and Department of Statistics Malaysia, respectively. Poisson generalised linear models were developed to assess the association between climatic parameters and the number of reported food poisoning cases. The findings revealed that the food poisoning incidence in Malaysia during the 11 years study period was 561 cases per 100 000 population for the whole country. Among the cases, females and the ethnic Malays most frequently experienced food poisoning with incidence rates of 313 cases per 100,000 and 438 cases per 100,000 population over the period of 11 years, respectively. Most of the cases occurred within the active age of 13 to 35 years old. Temperature gave a significant impact on the incidence of food poisoning cases in Selangor (95% CI: 1.033–1.479; $p$ = 0.020), Melaka (95% CI: 1.046–2.080; $p$ = 0.027), Kelantan (95% CI: 1.129–1.958; $p$ = 0.005), and Sabah (95% CI: 1.127–2.690; $p$ = 0.012) while rainfall was a protective factor in Terengganu (95% CI: 0.996–0.999; $p$ = 0.034) at lag 0 month. For a 1.0°C increase in temperature, the excess risk of food poisoning in each state can increase up to 74.1%, whereas for every 50 mm increase in rainfall, the risk of getting food poisoning decreased by almost 10%. The study concludes that climate does affect the distribution of food poisoning cases in Selangor, Melaka, Kelantan, Sabah, and Terengganu. Food poisoning cases in other states are not directly associated with temperature but related to monthly trends and seasonality.

data obtained was only for research purpose, but we did obtain the ethical approval for the research. Other data, such as monthly meteorological data, can be shared with others because we brought it from the Department of Meteorology, Malaysia. I have uploaded the meteorological data at this link: https://doi.org/10.6084/m9.figshare.20523684.v1" Morbidity data: Ministry of Health. The application must be registered under the NMRR at https://nmrr.gov.my/register. Once approved, we can use the NMRR approval to get the data from the Ministry of Health. Meteorological data: Meteorological data can be brought at this link https://www.met.gov.my/hubungi-kami/alamat-pejabat/ or email to Pusat Iklim Nasional, klim@met.gov.my Population data is accessible and can be obtained from this link https://pqi.stats.gov.my/search.php.

**Funding:** This research received financial assistance from IRAGS18-048-0049.

**Competing interests:** The authors have declared that no competing interests exist.

## Introduction

Food poisoning is defined as an acute onset of vomiting, diarrhoea, or other acute symptoms associated with the ingestion of compromised food or drinks [1]. It can also cause neurological symptoms such as muscle weakness, paraesthesia, paralysis, and even death. It is listed as one of the notifiable diseases and require mandatory notification under the Prevention and Control of Infectious Disease Act 1988 in Malaysia. Food poisoning is usually infectious or toxic and the causal agents can be pathogenic microbes, including bacteria (e.g. *Salmonella*, *E. coli*, and *Campylobacter*), viruses (e.g. Norovirus), parasites (e.g. *Ascaris*, *Cryptosporidium*, *Entamoeba histolytica* or *Giardia*), fungi (e.g. *Aspergillus*) or chemicals (e.g. mycotoxins, marine biotoxins, persistent organic pollutants, heavy metals). An outbreak of food poisoning is declared if there are two or more cases of a similar illness resulting from the ingestion of the same contaminated food or drink [2]. The typical foods that are mostly contaminated during the outbreaks are raw or undercooked poultry, dairy products, expired eggs, and vegetables and mostly from *Salmonella typhi*, *Staphylococcus aureus*, *Escherichia coli* and *Clostridium perfringens* [3]. Various factors may contribute to food poisoning like poor self-hygiene of food handlers, unavailability of clean water supply and unhygienic kitchen utensils or kitchen counters [3].

Food poisoning incidences in Malaysia showed an increasing trend over the past twenty years [4, 5]. In the 1980s, the incidences were below 10.2 cases per 100,000 population, but the incidences increased to more than 35.0 cases per 100,000 population in 1999, and consistently above 44.0 cases per 100,000 population after 2010 [3, 4]. The highest incidence recorded at the state level was in Melaka in 1996, with the incidence of 197.05 cases per 100,000 population, followed by 58.91 cases per 100,000 population in Kelantan in 2005 [4, 5]. The low incidence rate in the 1980s and 1990s compared to recent years are probably due to a lack of reported cases or registered cases.

Food poisoning remains a major public health problem worldwide. The distribution and incidence of food poisoning are influenced by fluctuations in weather patterns such as temperature, rainfall, and relative humidity as a result of climate change [6–9]. In a country that has a seasonal temperate climate such as Lebanon, it was proven that food poisoning illnesses peak during warmer months [6]. However, data from tropical countries like Malaysia where it is hot and humid throughout the year are limited to support the correlation of high food poisoning incidences in perpetually hot climate regions. Climate in Malaysia is influenced by the mountainous topography and complex land-sea interactions. The interannual variability of temperature in Malaysia is primarily dominated by the El Nino-Southern Oscillation (ENSO). Warm and dry climate provides favorable growth of pathogenic microbes, including those causing food-borne illnesses in the food supply chain [6, 7]. A warm climate can also affect human behaviours on food consumption and social interaction. People tend to drink more water, get more involved with recreational activities, and attend social gatherings like a wedding during the Southwest Monsoon (warmer months). These conditions increase the risk of food poisoning due to improper management in raw food supply, preparation, serving or during storage. Increased rainfall could also contribute to food poisoning as it leads to more water runoff, and increases the possibility of flooding which vehicles the pathogenic bacteria from the land such as human and animal excreta on the soil to the river or any drinking water sources, thus, increasing the risk for an outbreak of food poisoning.

Based on the Representative Concentration Pathways (RCPs) emissions, the scenarios indicated that the minimum increase of global mean surface temperature is likely to be from 0.3˚C to 1.7˚C under RCP 2.6, while the maximum increase is likely to be from 2.6˚C to 4.8˚C under RCP 8.5 at the end of the 21st century [10, 11]. In Malaysia, the observed surface temperature

data had also indicated an increase in warming between 0.32 and 1.5˚C for the last five decades [12, 13]. Peninsular Malaysia indicated a higher temperature increase compared to East Malaysia, and western Peninsular Malaysia indicated the highest compared to other states [12]. Malaysia also receives abundant rainfall annually and the East Coast areas experienced floods almost every year during the Northeast Monsoon (rainy season). With the increasing temperature, abundant rainfall during the Northeast Monsoon, and elevated food poisoning incidences, research on climatic factors influencing food poisoning incidences in hot tropical countries are essential. Previous research used Generalised linear Model (GLMs) [14–16], Generalized Additive Model (GAM) [17], Quasi-Poisson regression [18], and Pearson correlation regression [19] to examine the association between climate variables and infectious diseases, however GLMs are the most widely used. Therefore, this research aims to investigate the influence of climate variability on food poisoning cases in Malaysia using a Poisson generalised linear model. This research is important for future prediction of food poisoning in extreme climatic scenarios and taking into consideration the appropriate intervention measures that could be done in Malaysia.

## Materials and methods

### Study setting and study area

This is an ecological study that encompasses all 13 states in Malaysia. The study focuses on the Malaysian population rather than individuals. Any cases from Putrajaya and Kuala Lumpur were classified under Selangor state, while cases from Labuan were classified under Sabah state. This is to synchronise the cases with the local meteorological data because no meteorological stations are found in the areas stated.

### Study population

The study included the Malaysian population, irrespective of race, in all 13 states, including Sabah and Sarawak. The total population in Malaysia in 2014 was 30.72 million. Native Malays or Bumiputera recorded the highest number with 18.87 million (61.4%), followed by Chinese, Indians, and other races with 6.59 million (21.5%), 1.98 million (6.4%), and 0.27 million (0.9%), respectively. For non-Malaysians or immigrants, there were 3.01 million (9.8%) [20].

### Food poisoning data, population data, and climate data

In order to explore the relationship between climate and food poisoning cases in Malaysia, the monthly data of food poisoning cases where patients were diagnosed at the Ministry of Health (MOH) Malaysia and other healthcare facilities in Malaysia were collected from January 2004 to December 2014 from the MOH. For demographic data, basic information such as the locality or address, age, gender, race, date of onset, and date of analysis were recorded. All incomplete dataset without medical information such as the onset date, sampling date, and diagnosis date were rejected. All these food poisoning cases were converted to incidence data by dividing it with the population data obtained from the Department of Statistics Malaysia [20].

Monthly climate data such as relative humidity, temperature, and rainfall were obtained from the Malaysian Meteorological Department (MMD) for the years 2004 to 2014. The missing climate data were estimated using the multiple regression analysis (multiple imputation method) in SPSS. The collinearity between independent variables for each state was analysed prior to model development using the correlation test and Variance Inflation Factors (VIF). Data were analysed using IBM SPSS for Windows version 23. Descriptive analysis was used to

describe the demographic data, meteorological data, and morbidity data. A generalised linear model was applied for multivariate analyses.

The mean and maximum data for incidence of food poisoning, temperature and rainfall for the whole Malaysia were mapped using QGIS version 3.4 long-term release. The incidence of food poisoning used choropleth maps while rainfall used the inverse distance weighing (IDW) interpolation method. The classification used the quartile method from Q1 (the lowest) to Q4 (the highest).

## Model development

Poisson generalised linear models and overdispersed Poisson models estimating the understated standard errors were run to investigate the association between climate parameters and food poisoning incidence at lag 0 month and lag 1 month. The model equation for overdispersed Poisson was the same as that for Poisson regression. The overdispersion parameter did not affect the expected outcome, but it affected the estimated variance of the expected counts. Scale weight was used to control for overdispersion by dividing the deviance to 1. Fourier terms and time stratified model were included in the model to control for seasonality and the monthly trends, respectively. Seasonality refers to the presence of variations that occur at regular monthly intervals due to the weather factor. Trend refers to the direction of data, that is either increasing or decreasing, and linear assumption was used in this research. The analysis was considered towards pooled health impacts for the community at the state level. As the population increased over time, the population was set up as an offset.

The framework models were explained in the previous research [6, 15]. The framework was accommodated as expressed in Eqs (1) to (3):

$$Y_i \sim Pois(u_i \lambda_i) \tag{1}$$

$$log(u_i \lambda_i) = log(u_i) + log(\lambda_i) \tag{2}$$

$$log(\lambda_i) = X_i \beta \tag{3}$$

Where $u_i$ is the exposed population at that time; $\lambda_i$ is the expected number of cases as a fraction of the exposed population; $X_i$ is the regression matrix; and $\beta$ is the vector of model coefficients. The robustness of the model depends on the goodness of fit. The smallest Akaike information criterion (AIC) was chosen as the best model. Sensitivity testing was conducted to determine the sensitivity of significant independent variables such as temperature and rainfall in affecting food poisoning incidences. We used lag temperature or rainfall data to replace the climate variables (temperature or rainfall) at the Lag 0 data due to the unavailability of minimum and maximum climate data in this research. If the analysis is to ensure consistency of temperature in affecting food poisoning incidences, the lag rainfall data will be used and vice versa. Similar findings with the real model indicated the model was robust.

This research obtained ethical approval from the Faculty Medicine Universiti Kebangsaan Malaysia (FF-2013-424),National Medical Research Register (NMRR-13-1120-17533), and Medical Research and Ethics Committee (MREC), Ministry of Health Malaysia.

## Results

### Food poisoning cases in Malaysia

Tables 1 and 2 present the demographic information and annual incidence rate for food poisoning in Malaysia. The annual incidence rate by states ranged from 2.8 to 172.3 cases per

**Table 1. Sum of food poisoning incidences in Malaysia from 2004 to 2014.**

| Demographic Data | Food Poisoning Incidence (per 100 000 population) |
|---|---|
| **Gender** | |
| Male | 249 cases |
| Female | 313 cases |
| **Race** | |
| Malay | 438 cases |
| Non-Malay | 122 cases |
| **Age** | |
| Mean±SD | 20±18 years old |
| 0–12 years old | 174 cases |
| 13–35 years old | 331 cases |
| 36–58 years old | 27 cases |
| >59 years old | 5 cases |
| **Nationality** | |
| Malaysian | 551 cases |
| Non-Malaysian | 10 cases |
| **Death** | 2 cases (per 1 000 000 population) |

**Table 2. Distribution of annual incidence rates of food poisoning cases and climate in Malaysia from 2004–2014.**

| | Annual incidence rates from 2004–2014 (per 100,000 population) | | | | | Climate from 2004–2014 | | | |
|---|---|---|---|---|---|---|---|---|---|
| States | Min | Mean | Max | High incidence rates | Low incidence rates | High Temperature | Low Temperature | High Rainfall | Low Rainfall |
| Johor | 13.13 | 51.39 | 97.21 | May to June | November to December | May and June | December and January | December and January | February and June |
| Melaka | 21.67 | 67.22 | 113.64 | April to July | November to December | May and June | December and January | November and December | January and February |
| Negeri Sembilan | 5.99 | 45.42 | 135.52 | June to July | December | April and May | December and January | October and November | January and February |
| Selangor | 2.80 | 26.78 | 50.32 | April to June | November to December | June and July | November and December | October and November | January and February |
| Pahang | 38.03 | 70.03 | 132.80 | July to August | November to December | May and June | December and January | November and December | February and June |
| Terengganu | 6.62 | 90.69 | 160.55 | April, May, July | November to December | April and May | December and January | November and December | February and March |
| Kelantan | 28.96 | 90.05 | 155.89 | April to July | December to January | April and May | December and January | November and December | February and April |
| Perak | 14.08 | 58.65 | 116.04 | January, July to August | November to December | May and June | December and January | October and November | June and July |
| Penang | 8.07 | 41.72 | 136.14 | January, September to October | December | May and June | November and December | August, September, October | December, January, February |
| Kedah | 16.45 | 45.79 | 68.95 | February to April, September to October | November to December | Mac to May | November and December | August, September, October | January and February |
| Perlis | 8.68 | 82.88 | 172.30 | February to April, September to November | December | Mac to May | December and January | March, November, and December | January and February |
| Sabah | 3.15 | 36.89 | 58.11 | March to June | November to December | April and May | January and February | December and January | February and April |
| Sarawak | 9.81 | 38.05 | 58.65 | April to July | November to December | June and July | December and January | December and January | February, June and August |

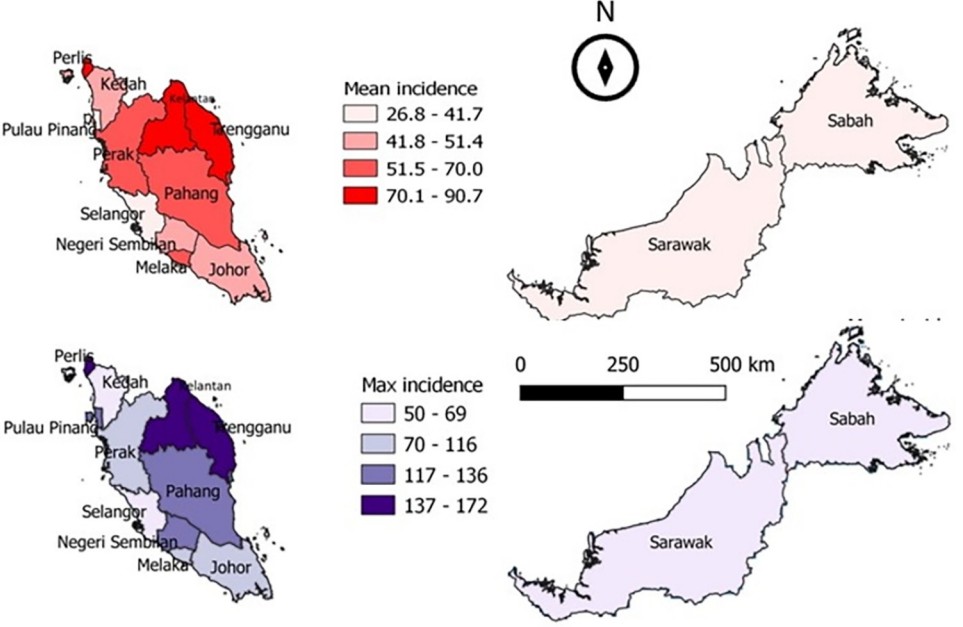

**Fig 1. Mean and maximum annual incidence rate of food poisoning in Malaysia per 100,000 population from 2004–2014.**

100,000 population while the total incidence of food poisoning for the whole country in Malaysia during the 11-year study period was 561 cases per 100,000 population. Generally, females and the ethnic Malays most frequently experienced food poisoning with incidence of 313 cases per 100, 000 and 438 cases per 100,000 population over the period of 11 years, respectively. Most of the cases occurred within the active age of 13 to 35 years old. There were some missing data in the demographic information, such as race and age, but less than 2% of the total cases. However, the data was not discarded as the main outcome in this research is the total number of population cases per month instead of individual information.

Large variations were found between the mean, and maximum incidence rates in each state as shown in Fig 1. Region-wise comparative data showed that the highest annual incidence rate was recorded in Perlis state, while the minimum annual incidence rate was in Selangor state with the incidence rate of 172 cases per 100,000 population and 3 cases per 100,000 population, respectively. High incidence rate in this research refers to the incidence rate above the 75[th] percentile, while low incidence rate refers to the incidence rate below the 25[th] percentile. Kelantan and Terengganu states were among the consistently highest incidence rates with the mean annual incidence rate of above 90 cases per 100,000 population. In the southern and central areas of Malaysia, Melaka consistently reported high incidence rates throughout the study period with more than 60 cases per 100,000 population. In the northern region, the incidence rates were high in Penang, Perak, and Perlis with more than 40 cases per 100,000 population. Most of the cases were recorded during the warmer months from April to August. However, states in the northern regions such as Kedah, Penang, Perak, and Perlis, had different trends compared to other states where there were two peak periods which were at the beginning of the year in January and February, and from September and October. The peak periods were at the end of the northeast monsoon and early inter-monsoon phase in Malaysia, respectively. All the states in Malaysia recorded the least food poisoning incidence rates during the northeast monsoon months in November and December.

## Climate variations in Malaysia

Data for the 11-year study period showed that the mean monthly temperature ranged from 24.7˚C to 30.0˚C (Fig 2). The data showed very small temperature variations as Malaysia only has southwest monsoon, northeast monsoon, and inter-monsoon between these two seasons. The maximum values of the mean monthly temperature attained from April to June coinciding with the inter-monsoon and southwest monsoon period with 50.0% of the recorded temperatures were 29.0˚C and above while the minimum values occurred in December or January in conjunction with the northeast monsoon period with 60% of the temperatures being below 25.0˚C (Table 2 and Fig 2). The trend test showed an upward trend of mean monthly temperature for almost all states from 2004–2014. However, the mean monthly temperature increases were small, below 0.5˚C in 2010–2014 compared to 2004–2009 ($p > 0.05$).

The rainfall data varied during the study period (Fig 3). The mean monthly rainfall and the maximum monthly rainfall for all states were between 150 mm/month– 300 mm/month, and 350 mm/month- 1250 mm/month, respectively. High rainfall was recorded during the inter-monsoon (October), and northeast monsoon, except for the northern region, whereby the trends differ as the region received high rainfall in March, and August to December. The east coast areas of Malaysia had rainfall of more than 1000 mm/month in November and December. Low rainfall was recorded at the end of the northeast monsoon and southwest monsoon for all states. The months which contributed the high rainfall tally with those months with low temperature as the data showed that temperature and rainfall were inversely correlated for almost all states ($r = -0.713$ to $-0.296$). The trend test showed an upward trend of rainfall for most states in 2004–2014 although the inter-annual changes were found to be insignificant ($p > 0.05$).

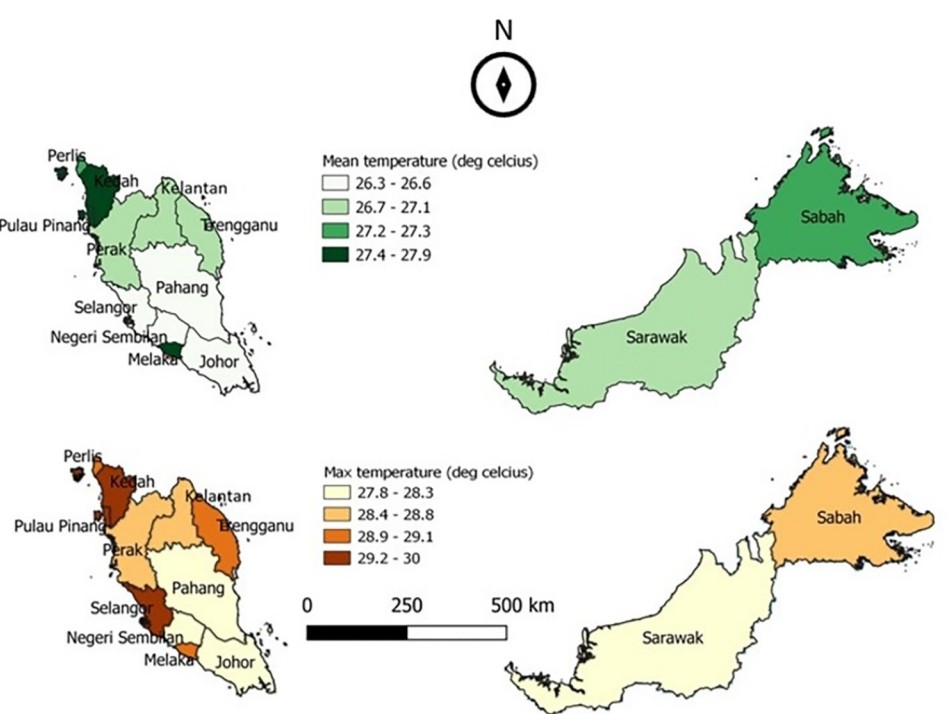

**Fig 2. Mean and maximum monthly temperature in Malaysia from 2004–2014 (˚C).**

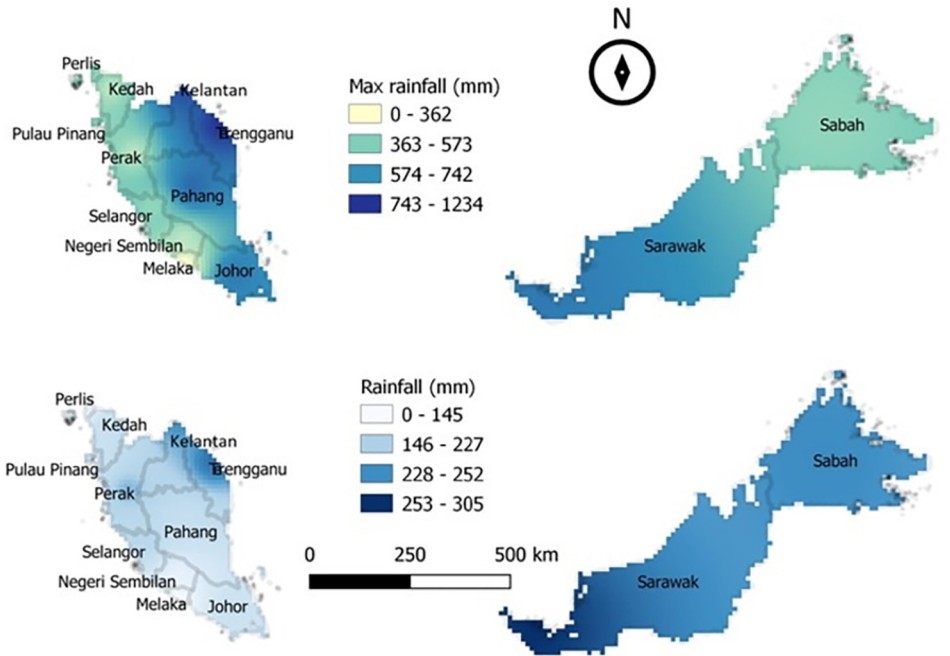

**Fig 3. Mean and maximum monthly rainfall in Malaysia from 2004–2014 (mm/month).**

## Association between climate and food poisoning incidence

Exploratory data analysis revealed a high level of multicollinearity between rainfall and relative humidity. As such, only one of these two factors can be used to ensure model stability. Due to limited information on relative humidity research related to food poisoning compared to rainfall and the abundant rainfall in Malaysia especially during the Northeast monsoon, relative humidity was excluded from the analysis.

The Poisson GLMs and overdispersed Poisson analyses were run for all states and the results showed that temperature was significantly related to food poisoning incidences in Selangor (95% CI: 1.033–1.479; $p = 0.020$), Kelantan (95% CI: 1.129–1.958; $p = 0.005$), Melaka (95% CI: 1.046–2.080; $p = 0.027$), and Sabah (95% CI: 1.127–2.690; $p = 0.012$) at lag 0 (Table 3 and Figs 4–7), and Kelantan (95% CI: 1.030 to 1.760; $p = 0.030$) at lag 1 month, while rainfall was significantly related to food poisoning incidences in Terengganu at lag 0 (95% CI: 0.996–0.999; $p = 0.034$) (Fig 8).

All food poisoning incidences in other states were associated only with seasonality and/or trend. The seasonality indicated that food poisoning rates were elevated mostly during the southwest monsoon and decreased during the northeast monsoon. The overall trend of food poisoning incidences for all states were slightly decreased over the 11 years of study period except for Negeri Sembilan and Perlis. For every 1.0°C increase in temperature, the excess risks of food poisoning in all states increased between 23.6% and 74.1%. The temperature was also tested at every increase of 0.25°C, 0.5°C and 0.75°C. The results showed an upward increment of excess risks of food poisoning as the temperature increases (Table 4). Rainfall was a protective factor in Terengganu. For every 1 mm increase in rainfall, the risk of getting food poisoning was decreased by less than 1%. Besides 1 mm, rainfall was also tested for every increase of 10 mm and 50 mm and the results showed, the risk can be decreased by almost 10% if the amount of rainfall was increased. The results were consistent with Table 2 as it

**Table 3.** Correlations between meteorological factors and food poisoning at lag 0 month (model 1) and Lag 1 month (model 2).

| | Model 1 | | | | Model 2 | | | |
|---|---|---|---|---|---|---|---|---|
| Variable | Model Value | Confidence Interval | Exp(B) | *p*-value | Model Value | Confidence Interval | Exp(B) | *p*-value |
| **Selangor**** | | | | | | | | |
| Temperature | 0.212 | 1.033–1.479 | 1.236 | 0.020* | 0.031 | 0.865–1.230 | 1.032 | 0.730 |
| Rainfall | 0.001 | 0.999–1.003 | 1.001 | 0.264 | -0.001 | 0.998–1.002 | 1.000 | 0.991 |
| Seasonality | 0.109 | 1.039–1.198 | 1.115 | 0.003* | 0.084 | 1.015–1.167 | 1.088 | 0.018* |
| Trend | -0.117 | 0.887–0.893 | 0.890 | <0.001* | -0.116 | 0.887–0.894 | 0.891 | <0.001* |
| **Kelantan*** * | | | | | | | | |
| Temperature | 0.396 | 1.129–1.958 | 1.487 | 0.005* | 0.297 | 1.030–1.760 | 1.346 | 0.030* |
| Rainfall | 0.001 | 0.999–1.002 | 1.000 | 0.533 | 0.001 | 0.998–1.002 | 1.000 | 0.819 |
| Seasonality | -0.012 | 0.854–1.143 | 0.988 | 0.874 | -0.101 | 0.778–1.050 | 0.904 | 0.188 |
| Trend | -0.020 | 0.975–0.985 | 0.980 | <0.001* | -0.020 | 0.976–0.985 | 0.980 | <0.001* |
| **Melaka**** | | | | | | | | |
| Temperature | 0.389 | 1.046–2.080 | 1.475 | 0.027* | 0.152 | 0.826–1.640 | 1.164 | 0.385 |
| Rainfall | -0.002 | 0.995–1.002 | 0.998 | 0.336 | 0.002 | 0.999–1.005 | 1.002 | 0.200 |
| Seasonality | -0.016 | 0.870–1.114 | 0.984 | 0.802 | -0.069 | 0.823–1.058 | 0.933 | 0.281 |
| Trend | -0.007 | 0.988–0.998 | 0.993 | 0.007* | -0.007 | 0.988–0.998 | 0.993 | 0.010* |
| **Sabah**** | | | | | | | | |
| Temperature | 0.555 | 1.127–2.690 | 1.741 | 0.012* | 0.353 | 0.932–2.173 | 1.423 | 0.103 |
| Rainfall | 0.002 | 0.999–1.004 | 1.002 | 0.191 | 0.002 | 1.000–1.005 | 1.002 | 0.191 |
| Seasonality | -0.037 | 0.865–1.073 | 0.963 | 0.497 | -0.091 | 0.820–1.017 | 0.913 | 0.097 |
| Trend | -0.060 | 0.938–0.946 | 0.942 | <0.001* | -0.060 | 0.938–0.946 | 0.942 | <0.001* |
| **Terengganu**** | | | | | | | | |
| Temperature | -0.244 | 0.560–1.094 | 0.783 | 0.152 | 0.225 | 0.888–1.766 | 1.252 | 0.200 |
| Rainfall | -0.002 | 0.996–0.999 | 0.998 | 0.034 | -0.001 | 0.998–1.001 | 0.999 | 0.272 |
| Seasonality | 0.113 | 0.980–1.279 | 1.120 | 0.095 | 0.085 | 0.948–1.250 | 1.089 | 0.228 |
| Trend | -0.007 | 0.987–0.998 | 0.993 | 0.006* | -0.011 | 0.984–0.995 | 0.989 | <0.001* |
| **Johor**** | | | | | | | | |
| Temperature | -0.131 | 0.546–1.343 | 0.877 | 0.546 | -0.071 | 0.631–1.374 | 0.931 | 0.720 |
| Rainfall | -0.002 | 0.996–1.001 | 0.998 | 0.133 | -0.001 | 0.997–1.001 | 0.999 | 0.555 |
| Seasonality | -0.034 | 0.860–1.086 | 0.967 | 0.568 | -0.054 | 0.850–1.056 | 0.948 | 0.331 |
| Trend | -0.034 | 0.962–0.971 | 0.967 | <0.001* | -0.035 | 0.961–0.971 | 0.966 | <0.001* |
| **Perak**** | | | | | | | | |
| Temperature | -0.231 | 0.577–1.092 | 0.794 | 0.156 | -0.016 | 0.706–1.371 | 0.984 | 0.923 |
| Rainfall | -0.001 | 0.997–1.001 | 0.999 | 0.484 | -0.001 | 0.996–1.001 | 0.999 | 0.283 |
| Seasonality | -0.066 | 0.854–1.026 | 0.936 | 0.159 | 0.046 | 0.949–1.155 | 1.047 | 0.361 |
| Trend | -0.013 | 0.983–0.992 | 0.987 | <0.001* | -0.014 | 0.982–0.991 | 0.986 | <0.001* |
| **Pahang** | | | | | | | | |
| Temperature | 0.102 | 0.851–1.441 | 1.107 | 0.448 | -0.168 | 0.648–1.104 | 0.846 | 0.217 |
| Rainfall | -0.001 | 0.996–1.002 | 0.999 | 0.434 | -0.002 | 0.996–1.001 | 0.998 | 0.247 |
| Seasonality | -0.077 | 0.831–1.032 | 0.926 | 0.162 | -0.089 | 0.823–1.017 | 0.915 | 0.099 |
| Trend | -0.006 | 0.990–0.999 | 0.994 | 0.012* | -0.006 | 0.990–0.999 | 0.994 | 0.011* |
| **Sarawak **** | | | | | | | | |
| Temperature | 0.269 | 0.881–1.943 | 1.309 | 0.182 | -0.049 | 0.644–1.409 | 0.953 | 0.808 |
| Rainfall | 0.001 | 0.999–1.002 | 1.001 | 0.483 | -0.001 | 0.998–1.001 | 1.000 | 0.824 |
| Seasonality | 0.247 | 1.031–1.590 | 1.280 | 0.025* | -0.099 | 0.832–0.987 | 0.906 | 0.024 |
| Trend | -0.109 | 0.822–0.979 | 0.897 | 0.015* | -0.027 | 0.970–0.978 | 0.974 | <0.001* |
| **Kedah**** | | | | | | | | |

*(Continued)*

**Table 3.** (Continued)

| | Model 1 | | | | Model 2 | | | |
|---|---|---|---|---|---|---|---|---|
| Variable | Model Value | Confidence Interval | Exp(B) | *p*-value | Model Value | Confidence Interval | Exp(B) | *p*-value |
| Temperature | -0.222 | 0.603–1.063 | 0.801 | 0.124 | -0.244 | 0.607–1.010 | 0.783 | 0.060 |
| Rainfall | 0.001 | 1.000–1.003 | 1.001 | 0.077 | 0.001 | 0.998–1.001 | 1.000 | 0.665 |
| Seasonality | -0.151 | 0.772–0.957 | 0.860 | 0.006* | -0.022 | 0.886–1.079 | 0.978 | 0.656 |
| Trend | -0.018 | -0.979–0.986 | 0.982 | <0.001* | -0.018 | 0.979–0.986 | 0.982 | <0.001* |
| **Penang**** | | | | | | | | |
| Temperature | -0.302 | 0.539–1.014 | 0.740 | 0.061 | -0.284 | 0.562–1.009 | 0.753 | 0.057 |
| Rainfall | 0.001 | 0.999–1.003 | 1.001 | 0.392 | 0.002 | 1.000–1.004 | 1.002 | 0.089 |
| Seasonality | -0.045 | 0.853–1.071 | 0.956 | 0.437 | -0.045 | 0.852–1.072 | 0.956 | 0.442 |
| Trend | -0.011 | 0.984–0.993 | 0.989 | <0.001* | -0.012 | 0.983–0.993 | 0.988 | <0.001* |
| **Negeri Sembilan** | | | | | | | | |
| Temperature | 0.110 | 0.790–1.577 | 1.116 | 0.534 | 0.211 | 0.905–1.684 | 1.235 | 0.183 |
| Rainfall | -0.002 | 0.996–1.001 | 0.998 | 0.198 | -0.002 | 0.996–1.001 | 0.998 | 0.162 |
| Seasonality | 0.051 | 0.936–1.183 | 1.052 | 0.395 | 0.039 | 0.925–1.168 | 1.040 | 0.511 |
| Trend | 0.009 | 1.003–1.015 | 1.009 | 0.003* | 0.009 | 1.003–1.014 | 1.009 | 0.003* |
| **Perlis** | | | | | | | | |
| Temperature | 0.121 | 0.742–1.717 | 1.129 | 0.572 | 0.054 | 0.735–1.516 | 1.005 | 0.771 |
| Rainfall | 0.001 | 0.998–1.004 | 1.001 | 0.688 | 0.001 | 0.997–1.003 | 1.000 | 0.813 |
| Seasonality | -0.074 | 0.773–1.115 | 0.929 | 0.429 | -0.091 | 0.771–1.081 | 0.913 | 0.289 |
| Trend | 0.005 | 0.998–1.012 | 1.005 | 0.164 | 0.005 | 0.998–1.012 | 1.005 | 0.166 |

**Overdispersed Poisson

*Significance at *p*<0.05

indicated the high incidence of food poisoning occurred during the southwest monsoon (warmer months) and low incidence in the northeast monsoon (rainy seasons). The deviance after controlling for over-dispersion was 1, and the Akaike's Information Criterion (AIC) was below 150 for all states. The final models were tested for sensitivity analysis, and the results indicated that the temperature and rainfall factors were still significant in Selangor, Kelantan, Melaka and Sabah, even if one of the independent variables was replaced.

## Discussion

The total food poisoning annual incidences in all states in Malaysia fluctuated and ranged from 3 to 172.0 cases per 100,000 population. The variations in the incidences were large, which is due to the denominator difference. Although Selangor was reported as the state with the highest food poisoning cases, the total population in Selangor was the highest in Malaysia, therefore lowering the incidence rates. The highest annual and monthly incidence rates were reported in the East Coast states such as Terengganu and Kelantan as it can reach up to 53.0 cases per 100,000 population in a month. Smaller states like Perlis and Melaka also indicated higher incidence rates due to lower population number.

Although all states were affected, only states that have a significant relationship with the climate: Selangor, Kelantan, Melaka, Sabah, and Terengganu, were explained in detail. Temperature alone gave a total percentage of more than 10.0% variation for food poisoning cases in Malaysia. The increase of 1˚C of temperature is associated with an excess risk of food poisoning in each state of between 23.6% to 74.1% for Lag 0 month, and 34.6% for Lag 1 month. The risk of getting food poisoning increased with increasing temperature, as shown in Table 4. The

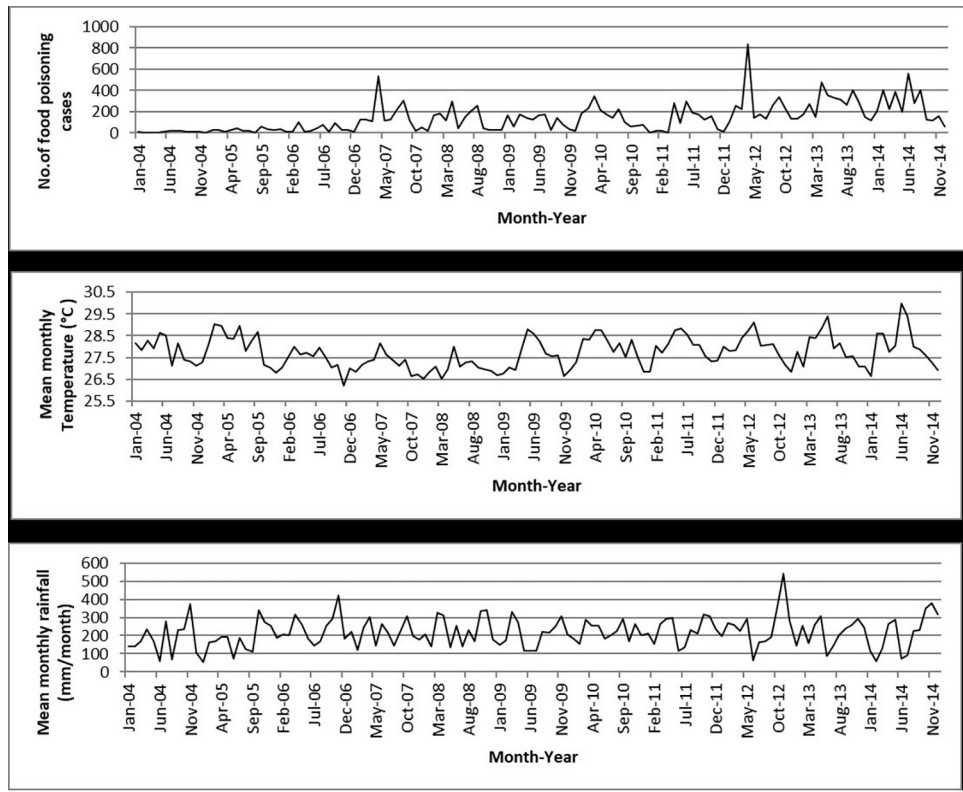

**Fig 4. Food poisoning cases and climate data in Selangor from 2004–2014.**

temperature in Kelantan was the only significant variable associated with an increased risk of food poisoning at Lag 1. This situation might be due to the acute effects and a short incubation period of food poisoning, and therefore, all other states showed insignificant findings. Most research showed lower excess risks compared to this research as the increase of 1.0˚C of temperature led to the percentage change of food poisoning cases of between 3.0% to 18.0% [6, 21, 22]. Malaysia's climate is hot and humid throughout the year compared to other studies that have four seasons. Most of the food poisoning cases investigated in other studies were related to cases of Salmonellosis food poisoning, while cases in Malaysia are cumulative cases of all causes of food poisoning. Variations in geography, culture, eating behaviour and technology, food type, and hygiene are also essential factors that can trigger food poisoning for example in Malaysia, the least cases were found among Chinese due to the preference to serve food hot from the wok [23–26]. Due to limited research specifically on the role of climatic variations on food poisoning, this research uses diarrhoeal research as the comparison. Zhou *et al*. [18] and Chou *et al*. [27] indicated a higher excess risk of getting food poisoning of 80.0% and between 20.0% to 52.0% depending on the age groups, respectively. Different studies recorded different excess risks and relative risks due to differences in terms of geographical area, climate variability, interpretation of enteric infections, method of analysis, and varying levels of temperature increase as some studies used an increase of 1.0, 2.0, and 5.0˚C [6, 7, 18, 27].

The research also found that rainfall was only significantly related to food poisoning in Terengganu with a risk reduction of almost 10% for every 50 mm increase of rainfall. Several studies supported this inverse relationship, but for diarrhoeal diseases [28–30]. Comparing the thresholds with other research was difficult because every area has different climates, rainfall

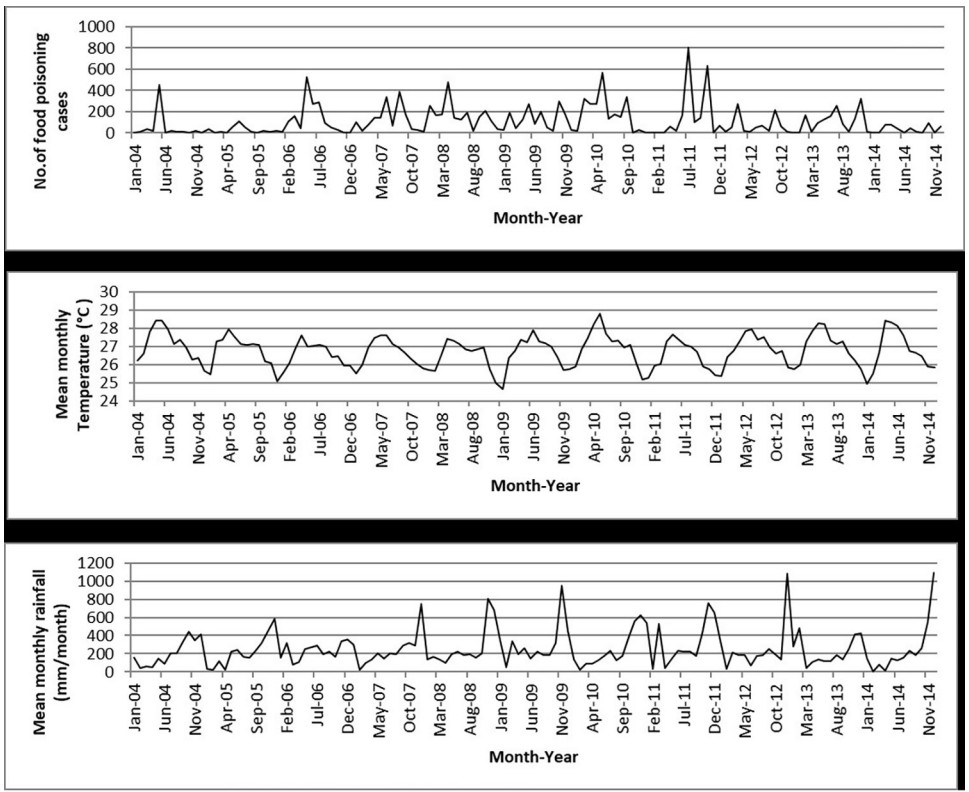

**Fig 5. Food poisoning cases and climate data in Kelantan from 2004–2014.**

intensities, cut-off points, and time-lags. Terengganu is one of the east coast states that receives abundant rainfall every year during the northeast monsoon season (November–December). During rainy days, people tend to stay at home and cook their own food. The food is usually consumed while it is still hot, therefore lessens the risk of getting food poisoning as there is control of the bacterial growth. In addition, rainfall could also protect the communities against exposure to contaminated recreational water bodies. All other states showed insignificant results which may be due to the monthly mean rainfall that were low in other areas. Rainfall is usually related to floods, and floods are very common in Malaysia. However, based on a litera-ture search conducted, very limited cases of food poisoning were reported during flood events in Malaysia. Badrul *et al.* [31] reported that only 46 cases of food poisoning occurred during the major floods in Johor in December 2006 and January 2007 whereas during the major flood in 2014, there were only 27 food poisoning cases reported in Pahang [32], and no case was reported in Kelantan [33]. Minimal cases of food poisoning during flood might be due to the complexity in specifically identifying food poisoning events from other enteric infections including gastroenteritis, or the successful public health interventions such as health education activities in the flood evacuation centres and through the media, improved sanitation and hygiene through the provision of safe water supply and toilets, and efficient management of the disposal of flood wastes and carcasses to prevent disease-breeding grounds.

Between February and July every year, many events, weddings, and family gatherings are usually organized during Chinese New Year, one-week school holidays in March, and the two-week mid-term holidays in May. The increased frequency of these events may elevate the potential of getting food poisoning, especially when the food is not prepared properly [34].

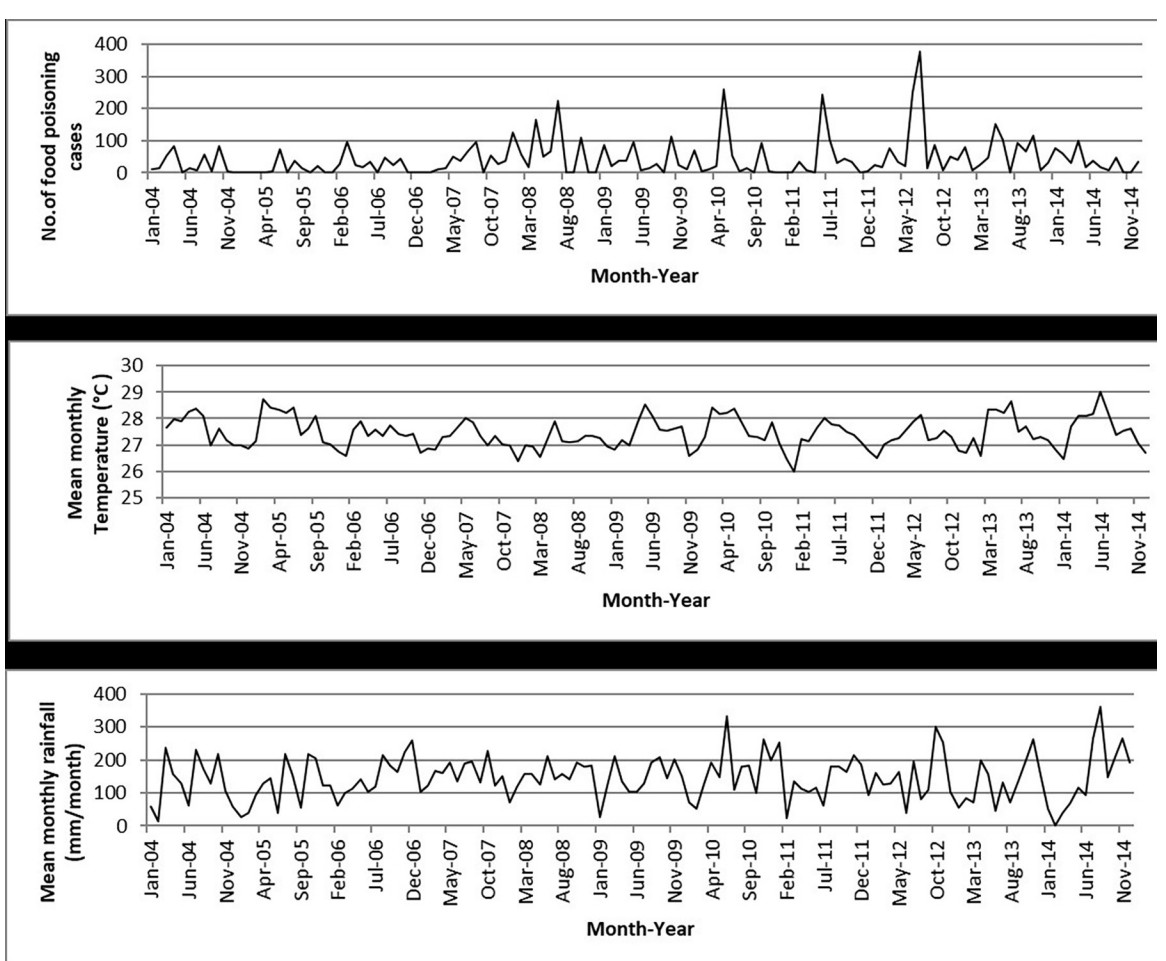

**Fig 6. Food poisoning cases and climate data in Melaka from 2004–2014.**

During the end of the northeast monsoon and southwest monsoon seasons, Malaysia receives a minimum amount of rainfall and therefore, the temperature was high during that period. Most of the food poisoning incidences were reported during the same period. Hot weather provides an optimal temperature range for the growth of many types of toxin-producing bacteria, such as *Staphylococcus aureus* (37.0˚C) [35], *Bacillus cereus* (30.0˚C to 37.0˚C) [36], and *Salmonella* (35˚C to 37˚C) [37]. These temperatures are within the maximum temperature range in Malaysia. Therefore, bacterial growth and toxin production are easily induced in foods that are not served at a sufficiently high temperature. The low incidence rates of food poisoning in November and December for all states may be related to comparatively low temperature and high rainfall.

The food poisoning incidences in Selangor, Melaka, Kelantan, Terengganu, and Sabah can be due to multiple factors. Selangor and Melaka received the highest temperature distributions after the northern areas. Selangor is also the most populous state in Malaysia, has a high number of squatters in pockets of the urban metropolitan area [38] which have unhygienic environment and lack of basic sanitation facilities. In addition, most of the communities in these areas are working and eating outside in restaurants. People can easily get infections through

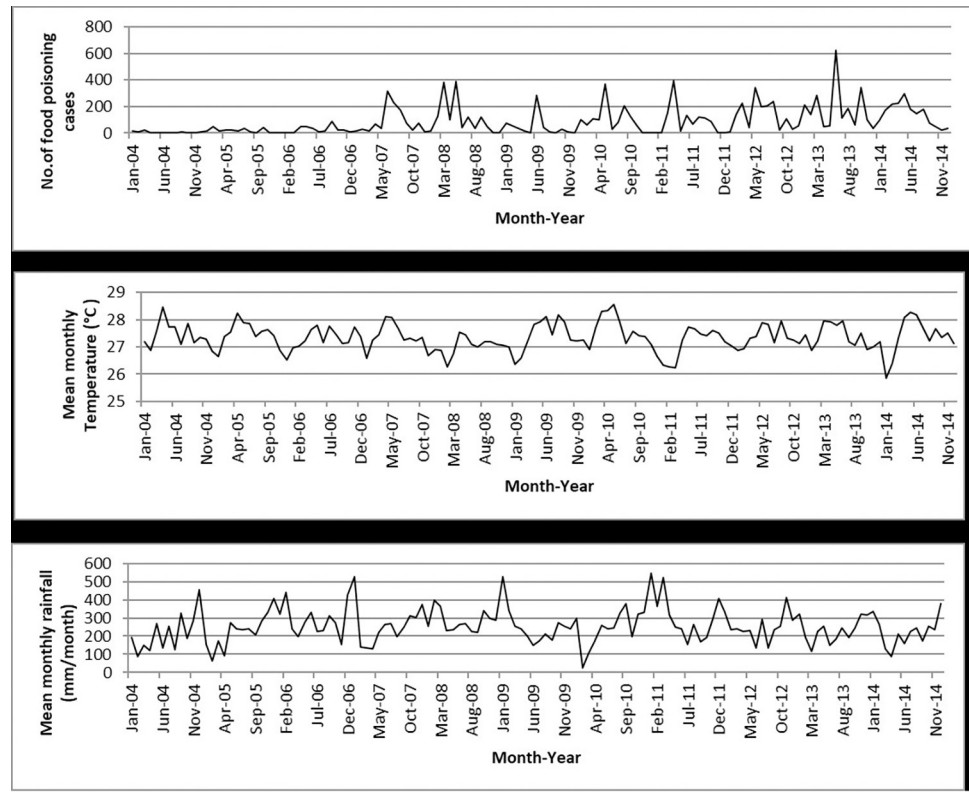

**Fig 7. Food poisoning cases and climate data in Sabah from 2004–2014.**

foods either during preparation, storage or serving [3, 24, 25] and the temperature can accelerate the food poisoning infection in these areas. The situation in Melaka is similar to Selangor where food contaminations most possibly occurred in food premises.

Food poisoning incidences in Terengganu, Kelantan and Sabah might have different possible contributing factors. Terengganu received the most abundant rainfall in Malaysia, followed by Kelantan and Pahang. However, rainfall was a protective factor for food poisoning only in Terengganu. In Kelantan and Sabah, high incidences of food poisoning were during the southwest monsoon due to hot weather. Apart from the climatic influences (temperature and rainfall), the large fraction of unexplained variance in food poisoning events observed in this area could be related to the prevalent disparity in the social factor.

The key strength of this study is that the retrospective data covered all the notifiable food poisoning cases for 11 years in all states in Malaysia. In addition, minimal publications are currently available on the impact of climate change on food poisoning cases in Malaysia and South East Asia. Therefore, it can be used as a reference. The limitation of this study is the limited climate stations in Malaysia; therefore, the data cannot be analysed at the district level. Moreover, food poisoning cases can be contributed by multiple factors; however, this research only focused on climate factors due to the limited parameters available for the retrospective data.

## Conclusion

In conclusion, climate does affect the distribution of food poisoning cases in Selangor, Melaka, Kelantan, Sabah, and Terengganu. For a 1.0°C increase in temperature, the excess risk of food

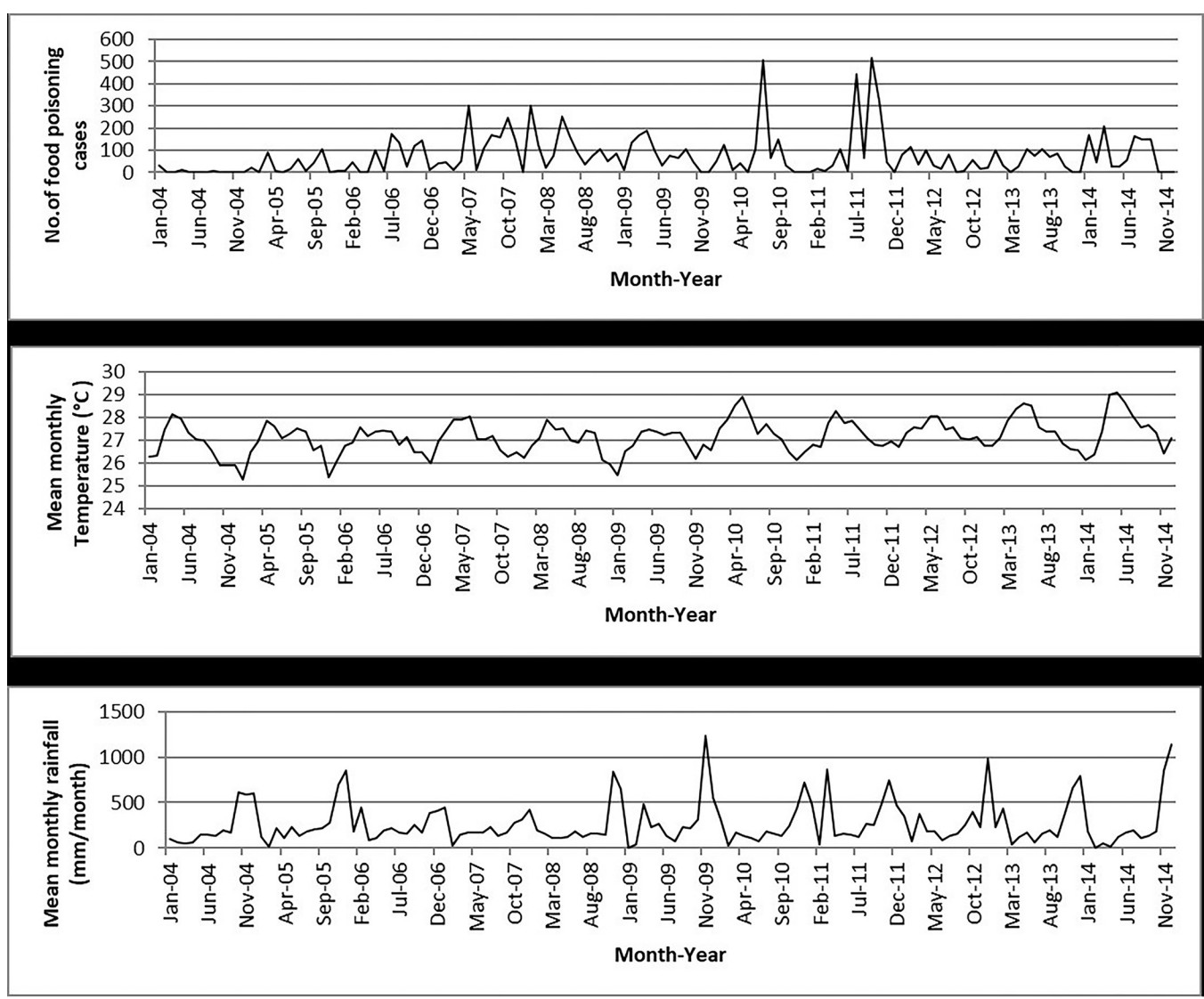

**Fig 8. Food poisoning cases and climate data in Terengganu from 2004–2014.**

poisoning in each state can increase up to 74.1%, whereas for every 50 mm increase in rainfall, the risk of getting food poisoning is decreased by almost 10%. The excess risk is directly associated with the hot climate in Selangor, Melaka, Kelantan, and Sabah, and inversely associated with rainfall in Terengganu. High rainfall in Terengganu is a protective factor for food poisoning. Food poisoning cases in other states are not directly associated with temperature but related to seasonality and/or trends. Different states might have additional different factors that contributed to the food poisoning. Besides the climate factors, different states have other multifactorial issues such as hygiene, sanitation, low coverage of clean water supply, food contamination in premises, and prevalent disparity in the social factor.

In the future, based on the RCP model worldwide, all countries will have at least a minimum increase in temperature from 0.3˚C to 1.7˚C, and Malaysia is not excluded. It will provide an optimum temperature for a lot of pathogenic organisms. Therefore, multidisciplinary approaches are needed to control food poisoning outbreaks in Malaysia. Some of the measures include

**Table 4. Comparison of excess risk at different temperatures and rainfall level.**

| Variable | B | Formula | Exp (B) | Excess Risk |
|---|---|---|---|---|
| **Selangor Lag 0** | | | | |
| Temperature increased by 0.25˚C | 0.212 | Exp (0.25*0.212) | 1.0544 | 5.4% |
| Temperature increased by 0.5˚C | 0.212 | Exp(0.5*0.212) | 1.111 | 11.1% |
| Temperature increased by 0.75˚C | 0.212 | Exp(0.75*0.212) | 1.172 | 17.2% |
| Temperature increased by 1˚C | 0.212 | Exp (0.212) | 1.236 | 23.6% |
| **Kelantan Lag 0** | | | | |
| Temperature increased by 0.25˚C | 0.396 | Exp (0.25*0.396) | 1.104 | 10.4% |
| Temperature increased by 0.5˚C | 0.396 | Exp (0.5*0.396) | 1.219 | 21.9% |
| Temperature increased by 0.75˚C | 0.396 | Exp(0.75*0.396) | 1.345 | 34.5% |
| Temperature increased by 1˚C | 0.396 | Exp (0.396) | 1.487 | 48.7% |
| **Kelantan Lag 1** | | | | |
| Temperature increased by 0.25˚C | 0.165 | Exp (0.25*0.165) | 1.042 | 4.2% |
| Temperature increased by 0.5˚C | 0.165 | Exp (0.5*0.165) | 1.085 | 8.5% |
| Temperature increased by 0.75˚C | 0.165 | Exp (0.75*0.165) | 1.132 | 13.2% |
| Temperature increased by 1˚C | 0.165 | Exp (0.165) | 1.179 | 17.9% |
| **Melaka Lag 0** | | | | |
| Temperature increased by 0.25˚C | 0.389 | Exp (0.25*0.389) | 1.102 | 10.2% |
| Temperature increased by 0.5˚C | 0.389 | Exp (0.5*0.389) | 1.214 | 21.4% |
| Temperature increased by 0.75˚C | 0.389 | Exp (0.75*0.389) | 1.338 | 33.8% |
| Temperature increased by 1˚C | 0.389 | Exp (0.389) | 1.475 | 47.5% |
| **Sabah Lag 0** | | | | |
| Temperature increased by 0.25˚C | 0.555 | Exp (0.25*0.555) | 1.148 | 14.8% |
| Temperature increased by 0.5˚C | 0.555 | Exp (0.5*0.555) | 1.319 | 31.9% |
| Temperature increased by 0.75˚C | 0.555 | Exp (0.75*0.555) | 1.516 | 51.6% |
| Temperature increased by 1˚C | 0.555 | Exp (0.555) | 1.741 | 74.1% |
| **Terengganu Lag 0** | | | | |
| Rainfall increased by 1mm | -0.002 | Exp (-0.002) | 0.998 | Decreased risk by 1% |
| Rainfall increased by 10mm | -0.002 | Exp (10*-0.002) | 0.980 | Decreased risk by 2% |
| Rainfall increased by 50mm | -0.002 | Exp (50*-0.002) | 0.905 | Decreased risk by 9% |

strengthening the management of water treatment, creating more awareness on food hygiene and sanitation, improving the surveillance system, inculcating the practices of hand washing with soap, and boiling water for at least one minute to kill the microbes. It is also advised to cook food at temperatures of more than 70.0˚C to kill all the dangerous microorganisms, and store foods at the safe temperature of below 4˚C. Besides controlling the temperature through boiling and freezing to avoid food spoilage, other standard methods to preserve foods are salting, sweetening, dehydration and canning. Finally, more inspection, food sampling, research activities and quality assurance programs are needed to reduce food poisoning cases in Malaysia.

## Acknowledgments

The authors would like to thank the Director-General of Health Malaysia for his permission to publish this article.

## Author Contributions

**Conceptualization:** Noor Artika Hassan, Jamal Hisham Hashim, Sharifa Ezat Wan Puteh, Shazlyn Milleana Shaharudin, Edre Mohammad Aidid.

**Data curation:** Noor Artika Hassan, Jamal Hisham Hashim, Wan Rozita Wan Mahiyuddin, Mohd Syazwan Faisal Mohd, Shazlyn Milleana Shaharudin.

**Formal analysis:** Noor Artika Hassan, Wan Rozita Wan Mahiyuddin, Shazlyn Milleana Shaharudin, Edre Mohammad Aidid, Isnizam Sapuan.

**Funding acquisition:** Noor Artika Hassan, Sharifa Ezat Wan Puteh.

**Investigation:** Noor Artika Hassan, Sharifa Ezat Wan Puteh.

**Methodology:** Noor Artika Hassan, Sharifa Ezat Wan Puteh, Wan Rozita Wan Mahiyuddin.

**Project administration:** Noor Artika Hassan, Jamal Hisham Hashim, Sharifa Ezat Wan Puteh.

**Resources:** Jamal Hisham Hashim, Mohd Syazwan Faisal Mohd.

**Software:** Mohd Syazwan Faisal Mohd, Shazlyn Milleana Shaharudin.

**Supervision:** Jamal Hisham Hashim, Sharifa Ezat Wan Puteh.

**Validation:** Noor Artika Hassan, Jamal Hisham Hashim, Sharifa Ezat Wan Puteh, Wan Rozita Wan Mahiyuddin, Mohd Syazwan Faisal Mohd, Shazlyn Milleana Shaharudin.

**Visualization:** Edre Mohammad Aidid, Isnizam Sapuan.

**Writing – original draft:** Noor Artika Hassan.

**Writing – review & editing:** Noor Artika Hassan.

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
