## [Editor Report · Decision Letter 0]

22 Dec 2021

PONE-D-21-31372

Climate Change and Temperature Rise: Implications on Food Poisoning Cases in Malaysia

PLOS ONE

Dear Dr. Hassan,

Thank you for submitting your manuscript to PLOS ONE. After careful consideration, we have decided that your manuscript does not meet our criteria for publication and must therefore be rejected.

I am sorry that we cannot be more positive on this occasion, but hope that you appreciate the reasons for this decision.

Yours sincerely,

Zaher Mundher Yaseen

Academic Editor

PLOS ONE

Additional Editor Comments:

Without any delays for the manuscript peer review. I am not able to find the appropriate reviewers to evaluate your research article.

- - - - -

---

## [Author Response · Author response to Decision Letter 0]

21 Jan 2022

Dear editor, 

Previously, the paper was rejected due to no reviewers to review the paper. I have suggested 3 names which are as follows :

1) Dr Sao Vibol, vibol.sao@rupp.edu.kh, from Cambodia and 

2) Prof Budi Haryanto from Indonesia, bharyanto@ui.ac.id

3)Associate Prof Dr Rozita Hod, rozita.hod@ppukm.ukm.edu.my from Malaysia

---

## [Decision Letter · Decision Letter 1]

19 Apr 2022

PONE-D-21-31372R1Climate Change and Temperature Rise: Implications on Food Poisoning Cases in MalaysiaPLOS ONE

Dear Dr. Noor Artika Hassan

Thank you for submitting your manuscript to PLOS ONE. After careful consideration, we feel that it has merit but does not fully meet PLOS ONE’s publication criteria as it currently stands. Therefore, we invite you to submit a revised version of the manuscript that addresses the points raised during the review process.

You are given 1 month to revise the commented manuscript. If you will need more time than this to complete your revisions, please reply to this message or contact the journal office at plosone@plos.org. Please include the following items when submitting your revised manuscript:A rebuttal letter that responds to each point raised by the academic editor and reviewer(s). You should upload this letter as a separate file labeled 'Response to Reviewers'.A marked-up copy of your manuscript that highlights changes made to the original version. You should upload this as a separate file labeled 'Revised Manuscript with Track Changes'.An unmarked version of your revised paper without tracked changes. You should upload this as a separate file labeled 'Manuscript'.

We look forward to receiving your revised manuscript.

Kind regards,

Mohamad Syazwan Mohd Sanusi

Academic Editor

PLOS ONE

Journal Requirements:

https://journals.plos.org/plosone/s/file?id=ba62/PLOSOne_formatting_sample_title_authors_affiliations.pdf”.

2. Please include your tables as part of your main manuscript and remove the individual files. Please note that supplementary tables (should remain/ be uploaded) as separate ""supporting information"" files

“This research received financial assistance from IRAGS18-048-0049.”

4.Thank you for stating the following in the Ethical approval Section of your manuscript:

“This research received financial assistance from IRAGS18-048-0049.”

“This research received financial assistance from IRAGS18-048-0049.”

7. Your ethics statement should only appear in the Methods section of your manuscript. If your ethics statement is written in any section besides the Methods, please delete it from any other section.

8. We note that Figures 1-6 in your submission contain [map/satellite] images which may be copyrighted. All PLOS content is published under the Creative Commons Attribution License (CC BY 4.0), which means that the manuscript, images, and Supporting Information files will be freely available online, and any third party is permitted to access, download, copy, distribute, and use these materials in any way, even commercially, with proper attribution. For these reasons, we cannot publish previously copyrighted maps or satellite images created using proprietary data, such as Google software (Google Maps, Street View, and Earth). For more information, see our copyright guidelines: http://journals.plos.org/plosone/s/licenses-and-copyright.

a. You may seek permission from the original copyright holder of Figures 1-6 to publish the content specifically under the CC BY 4.0 license.

9. We notice that your manuscript file was uploaded on Sep 29 2021. Please can you upload the latest version of your revised manuscript as the main article file, ensuring that does not contain any tracked changes or highlighting. This will be used in the production process if your manuscript is accepted. Please follow this link for more information: http://blogs.PLOS.org/everyone/2011/05/10/how-to-submit-your-revised-manuscript/

10. If possible, please upload a file showing your changes either highlighted or using track changes. This should be uploaded as a Revised Manuscript w/tracked changes, file type. Please follow this link for more information: http://blogs.PLOS.org/everyone/2011/05/10/how-to-submit-your-revised-manuscript/

11. Please include a separate legend for each figure in your manuscript.

Additional Editor Comments (if provided):

Reviewers' comments:

Reviewer's Responses to Questions

**Comments to the Author**

1. If the authors have adequately addressed your comments raised in a previous round of review and you feel that this manuscript is now acceptable for publication, you may indicate that here to bypass the “Comments to the Author” section, enter your conflict of interest statement in the “Confidential to Editor” section, and submit your "Accept" recommendation.

Reviewer #1: (No Response)

Reviewer #2: (No Response)

Reviewer #3: All comments have been addressed

2. Is the manuscript technically sound, and do the data support the conclusions?

Reviewer #1: Partly

Reviewer #2: Partly

Reviewer #3: Yes

3. Has the statistical analysis been performed appropriately and rigorously? 

Reviewer #1: Yes

Reviewer #2: Yes

Reviewer #3: Yes

4. Have the authors made all data underlying the findings in their manuscript fully available?

Reviewer #1: Yes

Reviewer #2: (No Response)

Reviewer #3: Yes

5. Is the manuscript presented in an intelligible fashion and written in standard English?

Reviewer #1: Yes

Reviewer #2: (No Response)

Reviewer #3: Yes

6. Review Comments to the Author

Reviewer #1: An interesting topic on climate change and its health implication on the population. However, kindly resolve the following critiques

1. See text highlighted in yellow in the manuscript with associated note attached

2. In the figures: the Maps show only 12 states representation, Kelantan was not depicted

3. In the discussion section: Reference was made to table 5 which was not available in the manuscript

4. conclusion: I observed that the recommendations made were all geared towards food safety with no recommendation focusing on prevention or easing of climate change impacts on food preservation to militate against food poisoning

Reviewer #2: Title: Climate Change and Temperature Rise: Implications on Food Poisoning Cases in Malaysia

Manuscript Number: PONE-D-21-31372R1

Comments to the Authors

Dear Authors,

The title is interesting and has public health importance in preventing food poisoning. To make it more interesting, please consider my comments below.

Abstract

1. On line number 27, what does it mean “…..the food poisoning incidence proportion……”? Please restate the statement. Could you remove the word “proportion”, please? In addition, make the change consistent throughout the entire document.

2. On line numbers 38 & 39, what is your evidence to conclude as “Food poisoning cases in other 39 states are not directly associated with temperature but related to monthly trends and seasonality”.? What are the factors that make the incidence of food poisoning seasonal?

Materials and Methods

1. On line number 105, please say “Materials and Methods” instead of “Methodology”.

2. Where was the study conducted? Please incorporate the study area and setting.

3. What type of study design have you used?

4. On line numbers 154 & 155, please rewrite the statement “The smallest deviance, 155 Pearson Chi Square, and Akaike information criterion (AIC) were chosen as the best model”.

Results

1. On line numbers 179-181, what is your evidence to say “High incidence rate refers to the incidence rate above the 75th percentile, while low incidence rate refers to the incidence rate below the 25th 180 181 percentile”?

Discussion

1. On line numbers 246-250, why did you incorporate it as the first paragraph of the discussion section? It would be better if you put the summary of the objective, method, and main findings of the study.

Conclusions

1. Please incorporate a short and informative conclusion under the heading “Conclusion”. It is not part of the “Discussion” section.

References

1. Please cite using the recommended citation style by PLoS ONE journal. Why do you use the Harvard citation style?

2. Reference numbers 24, 25, & 31 are arranged in Vancouver citation style. But the rest are arranged in Harvard citation style. Please don’t use a mix-up reference style.

Figure and table captions

1. Figure and table captions should convey a complete message. Please revise them.

2. Under table 3, please replace the word “coefficient” with “Variable”.

Reviewer #3: Climate Change and Temperature Rise: Implications on Food Poisoning Cases in Malaysia

General

This paper presents findings on an important area of research is in line with the SDGs specific to climate change. Additionally, food poisoning is an important public challenge which has been shown to depend on climate change. The use of retrospective data also provides a large sample size to improve the power of the study without missing the influence of inherent bias of retrospective studies.

Introduction

No comments

Methods

Authors reported on missing of some demographic characteristics. Authors failed to mention the numbers and proportions that were missing. This should be reported in the methods section for clarity.

In the development and execution of GLM, did authors adjust or control for any variables that may affect incidence of food poisoning. If so, can they add this to the methods section, if not then authors should discuss this as a limitation in the discussion.

Results

In table 3, 95% percent confidence intervals were reported, however, none have been used to support the results reported with the p-values. I suggest authors add these the main results section and the results section of the abstract.

Discussion

Authors should discuss the key strength of this study and the inherent limitations of using a retrospective data specific to this study.

Conclusion

No comments

7. PLOS authors have the option to publish the peer review history of their article (what does this mean?). If published, this will include your full peer review and any attached files.

Reviewer #1: No

Reviewer #2: No

Reviewer #3: **Yes: **DAVID NANA ADJEI

---

## [Author Response · Author response to Decision Letter 1]

11 Oct 2022

All comments have been addressed and I hope it will be given full consideration for publication

---

## [Editor Report · Decision Letter 2]

29 Nov 2022

PONE-D-21-31372R2Climate Change and Temperature Rise: Implications on Food Poisoning Cases in MalaysiaPLOS ONE

Dear Dr. Hassan,

Thank you for submitting your manuscript to PLOS ONE. After careful consideration, we feel that it has merit but does not fully meet PLOS ONE’s publication criteria as it currently stands. Therefore, we invite you to submit a revised version of the manuscript that addresses the points raised during the review process.

We look forward to receiving your revised manuscript.

Kind regards,

Mohamad Syazwan Mohd Sanusi

Academic Editor

PLOS ONE
---

## [Author Response · Author response to Decision Letter 2]

18 Jan 2023

I have done my very best to make all the corrections. I hope that this manuscript will be considered for publication in PLOS ONE. I would like to thank the reviewers and the academic editor for the positive feedback and valuable suggestions to improve this manuscript

---

## [Editor Report · Decision Letter 3]

30 Jan 2023

PONE-D-21-31372R3Investigation of the impacts of climate change and rising temperature on food poisoning cases in MalaysiaPLOS ONE

Dear Dr. Noor,

Thank you for submitting your manuscript to PLOS ONE. After careful consideration, we feel that it has merit but does not fully meet PLOS ONE’s publication criteria as it currently stands. Therefore, we invite you to submit a revised version of the manuscript that addresses the points raised during the review process.Please ensure that your decision is justified on PLOS ONE’s publication criteria and not, for example, on novelty or perceived impact.

We look forward to receiving your revised manuscript.

Kind regards,

Mohamad Syazwan Mohd Sanusi

Academic Editor

PLOS ONE

Journal Requirements:

Additional Editor Comments (if provided):

ACADEMIC EDITOR COMMENTS:

Dear author, kindly check the compile Fig. 1-6 that I've made. Please consider other revision eg., caption, text body etc. for the Figures in the main manuscripts.

Minor error:

Line 56: Foods that are most commonly contaminated - passive voice. The typical foods that are mostly contaminated during the outbreaks....

Line 71: space

Line 79: space

Line 160-162: symbol: italic, subscript etc.

Line 207: inter-monsoon phase?

Line 225: italic p? check throughout text body

Line 233: The east coast areas

Line 237: italic r? Pearson correlation R?

Table 3: to indicate a range, you need to use" - " instead of "to"
---

## [Author Response · Author response to Decision Letter 3]

15 Feb 2023

This manuscript has been revised based on the comments from the editors, reviewers, and academic editor.

These are responses based on revision 3 (comments from an academic editor).

Comment 1: Kindly check the compile Fig. 1-6 that I've made. Please consider other revision eg., caption, text body etc. for the Figures in the main manuscripts.

Feedback: I have edited the sentences in lines 198, 215, 219, 225, 228-229, 240,251,253-258. Thank you for the compilation of the figures. I truly appreciated your help.

Comment 2: Line 56: Foods that are most commonly contaminated - passive voice. The typical foods that are mostly contaminated during the outbreaks....

Feedback: I have edited the sentence in lines 56-57

Comment 3 and 4: Line 71 and 79 : space

Feedback: I have edited and checked the whole paragraph from lines 70-88.

Comment 5: Line 160-162: symbol: italic, subscript etc.

Feedback: I have edited the symbols as suggested, italic and subscript from lines 158-160.

Comment 6: Line 207: inter-monsoon phase?

Feedback: I have edited the words in line 212.

Comment 7: Line 225: italic p? check throughout text body

Feedback: I have made changes in lines 34-37, 227, 239,249-253,278, 280. 

Comment 8: Line 233: The east coast areas

Feedback: The sentence has been edited in lines 233-234.

Comment 9: Line 237: italic r? Pearson correlation R?

Feedback: The sentence has been edited in line 238.

Comment 10: Table 3: to indicate a range, you need to use" - " instead of "to"

Feedback: The word to has been replaced to - in Table 3 in line 278.

---

## [Editor Report · Decision Letter 4]

3 Mar 2023

Investigation of the impacts of climate change and rising temperature on food poisoning cases in Malaysia

PONE-D-21-31372R4

Dear Dr. Artika,

We’re pleased to inform you that your manuscript has been judged scientifically suitable for publication and will be formally accepted for publication once it meets all outstanding technical requirements.

Kind regards,

Mohamad Syazwan Mohd Sanusi

Academic Editor

PLOS ONE

---

## [Editor Report · Acceptance letter]

13 Mar 2023

PONE-D-21-31372R4 

Investigation of the impacts of climate change and rising temperature on food poisoning cases in Malaysia 

Dear Dr. Hassan:

I'm pleased to inform you that your manuscript has been deemed suitable for publication in PLOS ONE. Congratulations! Your manuscript is now with our production department. 

Kind regards, 

on behalf of

Dr. Mohamad Syazwan Mohd Sanusi 

Academic Editor

PLOS ONE